# Programmed Death 1 Ligand Expression in the Monocytes of Patients with Hepatocellular Carcinoma Depends on Tumor Progression

**DOI:** 10.3390/cancers12082286

**Published:** 2020-08-14

**Authors:** Akira Asai, Hidetaka Yasuoka, Masahiro Matsui, Yusuke Tsuchimoto, Shinya Fukunishi, Kazuhide Higuchi

**Affiliations:** The Second Department of Internal Medicine, Osaka Medical College, Takatsuki 5698686, Japan; yh0403.4351@gmail.com (H.Y.); masa1987_11_18@yahoo.co.jp (M.M.); in2141@osaka-med.ac.jp (Y.T.); in2104@osaka-med.ac.jp (S.F.); higuchi@osaka-med.ac.jp (K.H.)

**Keywords:** CD14^+^ cells, hepatocellular carcinoma, programmed death 1 ligands

## Abstract

Monocytes (CD14^+^ cells) from advanced-stage hepatocellular carcinoma (HCC) patients express programmed death 1 ligand (PD-L)/PD-1 and suppress the host antitumor immune response. However, it is unclear whether cancer progression is associated with CD14^+^ cells. We compared CD14^+^ cell properties before and after cancer progression in the same HCC patients and examined their role in antitumor immunity. CD14^+^ cells were isolated from 15 naïve early-stage HCC patients before treatment initiation and after cancer progression to advanced stages. Although CD14^+^ cells from patients at early HCC stages exhibited antitumor activity in humanized murine chimera, CD14^+^ cells from the same patients after progression to advanced stages lacked this activity. Moreover, CD14^+^ cells from early HCC stages scantly expressed PD-L1 and PD-L2 and produced few cytokines, while CD14^+^ cells from advanced stages showed increased PD-L expression and produced IL-10 and CCL1. CD14^+^ cells were also isolated from five naïve advanced-stage HCC patients before treatment as well as after treatment-induced tumor regression. The CD14^+^ cells from patients with advanced-stage HCC expressed PD-L expressions, produced IL-10 and CCL1, and exhibited minimal tumoricidal activity. After treatment-induced tumor regression, CD14^+^ cells from the same patients did not express PD-Ls, failed to produce cytokines, and recovered tumoricidal activity. These results indicate that PD-L expression as well as CD14^+^ cell phenotype depend on the tumor stage in HCC patients. PD-L expressions of monocytes may be used as a new marker in the classification of cancer progression in HCC.

## 1. Introduction

Hepatocellular carcinoma (HCC) is the fourth most common cause of cancer-related death worldwide. The World Health Organization estimates that more than 1 million individuals will die from HCC in 2030 [1]. The majority of cases occur in patients with liver disease, mostly as a result of hepatitis B or C virus (HBV or HCV) infection, alcohol abuse, or nonalcoholic steatohepatitis. The five-year survival rate of HCC patients is only 18%, making it the second most lethal tumor [2]. Therapeutic options are primarily selected on the basis of tumor stage and the extent of liver dysfunction. In the past, HCC was treated by surgical resection or radiofrequency ablation. However, frequent synchronous or metachronous recurrence in the form of new tumors or intrahepatic metastases led to high mortality rates in HCC patients [3].

Recently, immune checkpoint inhibitors have become available as a new treatment option for advanced-stage HCC [4]. These compounds target the programmed death 1 (PD-1)/programmed death ligand (PD-L) axis, which is involved in cancer immune escape or evasion [5]. PD-L1, which is expressed in HCC tumor cells, interacts with PD-1 receptors on activated T cells, leading to their inactivation [6,7,8] and ultimately suppressing the antitumor immune response of effector cells [9]. However, it has been reported that some PD-L1-positive tumors do not respond to the anti-PD-1 antibody, while a proportion of PD-L1-negative tumors do [10,11]. These discrepancies are not fully understood, although several mechanisms have been proposed [12,13].

The tumor microenvironment plays an important role in the establishment and progression of tumors. Among the immune cells in the HCC tissue, tumor-associated macrophages (Mϕ, TAMs) sustain tumor progression and are recruited from circulating CD14^+^ cells (monocytes) to the tumor microenvironment through tumor-derived signals [14]. In response to microbial stimuli and IFN-γ, M1 polarization occurs, thus leading to tumoricidal activity by producing high amounts of toxic intermediates. However, once tumors are progressed, the Mϕ that infiltrate the tumor tissue differentiate into M2Mϕ, which promote growth, invasion, and metastasis of tumor cells, thus inducing angiogenesis and suppressing antitumor immunity. TAMs often play a central role in tumor progression, and many TAMs have the property of M2Mϕ. Three different M2Mϕ phenotypes (M2Maϕ, M2bMϕ, and M2cMϕ) have been described [15] that are distinguished from each other based on their gene expression profiles, chemokine production, and surface marker expression [16,17]. Specifically, IL-10- and CCL17-producing Mϕ with mannose receptor gene expression are identified as M2aMϕ, IL-10- and CCL1-producing Mϕ are classified as M2bMϕ, and IL-10- and CXCL13-producing Mϕ with mannose receptor gene expression are recognized as M2cMϕ.

In a previous study, CD14^+^ cells detected in peripheral blood of patients with advanced stages of HCC were characterized as the M2b phenotype and were found to be a significant contributor to tumor growth promotion [18]. Meanwhile, in our recent report, we showed that the monocytes of patients with advanced-stage HCC expressed PD-L1 and PD-L2 and suppressed the antitumor immune response of other effector cells. Notably, these patients had a poor prognosis [19]. However, the impact of these cells on the host immune response against HCC is unknown. Moreover, it is also unclear whether these cells are induced by cancer progression or whether they actively contribute to this process. In this study, we investigated the relationship between disease progression and PD-L expression in monocytes of HCC patients. Finally, the tumoricidal activity of these monocytes against HCC was examined.

## 2. Results

### 2.1. Growth of HepG2 Cell-Derived Tumors in Chimeric Mice with CD14^+^ Cells from HCC Patients

To compare the antitumor activity of CD14^+^ cells from HCC patients at different tumor stages, xNSG mice (NSG mice exposed to whole-body X-ray radiation) were used to generate humanized murine chimeras. Specifically, the mice were injected with HepG2 cells and then with CD14^+^ cells from the same patients at early tumor stages (at the time of initial treatment) or at advanced tumor stages (Figure 1). The growth of solid tumors was measured weekly for six weeks. At the end of this period, the size of tumors in xNSG mice not transplanted with CD14^+^ cells was 253 ± 93.9 mm^3^. Notably, tumor growth was not detected in xNSG mice transplanted with CD14^+^ cells from early-stage HCC patients. However, solid tumors were observed in xNSG mice that had been transplanted with CD14^+^ cells collected at advanced tumor stages (229 ± 88.9 mm^3^). These results indicate that the antitumor activity of CD14^+^ cells in the same patients changed with tumor progression.

### 2.2. Differences in the Properties of CD14^+^ Cells in the Same Patients with Early and Advanced HCC

Tumor stage-related changes in the properties of CD14^+^ cells were examined. CD14^+^ cells were stained for PD-L1 and PD-L2 (Figure 2a,b). CD14^+^ cells isolated from patients with early-stage HCC expressed PD-L1 (41.2 ± 9.7%) and PD-L2 (24.5 ± 4.5%). However, the expression of these PD-Ls was higher in CD14^+^ cells obtained from the same patients at advanced HCC stages (PD-L1: 50.0 ± 7.8%, PD-L2: 36.2 ± 3.9%). Further, CD14^+^ cells (1 × 10^6^ cells/mL) from the same patients with early stages and advanced stages were cultured for 48 h, and the culture media was assayed by ELISA for production of IL-12, IL-10, CCL17, CCL1, and CXCL13 (Figure 2c). IL-12 was not produced by either population of CD14^+^ cells. However, IL-10 production was higher in CD14^+^ cells from advanced-stage HCC patients (212.9 ± 146.9 pg/mL) than in those from early-stage patients (64.5 ± 49.6 pg/mL). The production levels of CCL17 and CXCL13 did not differ between the two groups of CD14^+^ cells. CCL1 production was significantly higher at advanced HCC stages (2.84 ± 3.49 ng/mL) compared to early tumor stages (0.03 ± 0.08 ng/mL). In conclusion, CD14^+^ cells from patients with early-stage HCC exhibited poor PD-L, IL-12, and IL-10 expression and were negative for CCL17, CCL1, and CXCL13 (considered a quiescent Mϕ phenotype). Meanwhile, CD14^+^ cells from the same patients at advanced HCC stages acquired PD-L1 and PD-L2 expression and exhibited a IL-12^−^IL-10^+^CCL17^−^CCL1^+^CXCL13^−^ phenotype (considered a M2bMϕ phenotype).

### 2.3. Tumor Regression Is Associated with Restoration of CD14^+^ Cell Properties

In a small subset of patients with advanced-stage HCC, the treatments induced tumor regression. CD14^+^ cells isolated before treatment initiation from these responsive patients exhibited high expression of both PD-L1 (84.1 ± 15.7%) and PD-L2 (81.7 ± 13.9%). Notably, in CD14^+^ cells collected from the same patients with early stages of HCC after initial treatment, expression levels of both PD-Ls were found to be decreased (PD-L1, 18.7 ± 4.5%; PD-L2, 50.3 ± 35.7%) (Figure 3a). Moreover, although IL-10 (189 ± 78 pg/mL) and CCL1 (3.2 ± 1.9 ng/mL) were detected in the culture media of CD14^+^ cells from patients with advanced disease, the production of these cytokines was abolished after treatment and tumor regression (Figure 3b).

We also investigated PD-L expression in CD14^+^ cells from the peripheral blood of an additional 89 HCC patients with various stages of HCC. CD14^+^ cells from patients with early stages of HCC had weak expression of PD-L1 or PD-L2. Meanwhile, CD14^+^ cells from advanced stages strongly expressed both PD-Ls (Figure 4). Subsequently, the cytotoxic effect of CD14^+^ cells against HepG2 cells was examined. CD14^+^ cells exhibited weaker tumoricidal activity in patients with advanced HCC compared to those with early-stage HCC. Moreover, treatment-induced tumor regression was associated with the restoration of antitumor activity in CD14^+^ cells. These results indicate that the properties of CD14^+^ cells depend on the state of HCC progression (Figure 5).

## 3. Discussion

In this study, the antitumor activity of monocytes was investigated in relation to cancer progression. Monocytes are innate immune cells that serve as important regulators of cancer development and progression. These cells appear to play a dichotomous role depending on the cancer type/tissue of origin as well as the tumor microenvironment and stage of tumor growth [20]. CCL2, produced by tumor cells and associated stromal cells, is one of the best characterized tumor-derived factors that induces chemotaxis in monocytes, causing circulating monocytes to be recruited from the peripheral blood into the tumor sites [14,21]. During early stages of tumor growth, recruited monocytes directly kill tumor cells via cytokine-mediated induction of cell death and phagocytosis (M1Mϕ). Specifically, the recruited monocytes exposed to IFN-γ produce tumor necrosis factor-related apoptosis-inducing ligand (TRAIL), which induces cell death in TRAIL-sensitive tumor cells and stimulates secretion of CCL2 and IL-8 from tumor cells. Meanwhile, tumors that manage to escape immune surveillance mechanisms will progress, and M1Mϕ cells will become reprogrammed within the tumor environment to limit their cytotoxicity and differentiate into TAMs (M2Mϕ) [22]. These cells then produce IL-10 and TGF-β, which function to suppress the activities of other antitumor effector immune cells [15,23]. In this study, we investigated the phenotype of CD14^+^ cells (monocytes) from the peripheral blood before they were recruited to the tissues.

Results showed that CD14^+^ cells from patients with early stages of HCC were PD-L1^−^PD-L2^−^IL-12^−^IL-10^−^CCL17^−^CCL1^−^CXCL13^−^ (considered as a quiescent phenotype), while CD14^+^ cells from the same patients after tumor progression were PD-L1^+^PD-L2^+^IL-12^−^IL-10^+^CCL17^−^CCL1^+^CXCL13^−^ (considered to be the M2b phenotype). These results suggest that circulating monocytes from patients with advanced stages of HCC had already become skewed toward the M2b phenotype in the peripheral blood by the influence of tumor-associating factors. When these monocytes then become recruited to the tumor tissues by CCL2, they facilitate tumorigenesis by promoting immune suppression, extracellular matrix remodeling, angiogenesis, and tumor cell intravasation into the vasculature. Similarly, in other cancers, it has been reported that properties of peripheral blood monocytes are associated with better survival [24,25,26]. These results suggest that discrimination of monocyte phenotypes may provide a diagnostic or prognostic marker for HCC. Furthermore, the development of treatments that target monocyte differentiation may prove effective. 

The generation of these M2b monocytes/Mϕ during cancer progression could be affected by immune complex formation as various factors capable of inducing M2bMϕ have been described, including immune complexes [27,28]. Moreover, immune complexes targeting cancer antigens have been detected in the serum of patients with various cancers [29,30]. Therefore, it is reasonable to postulate that as cancer progresses, serum immune complexes increase, promoting the production of M2b monocytes from quiescent monocytes. It is further speculated that the M2b monocytes then suppress antitumor effector cells by producing cytokines, thereby promoting cancer progression.

Another mechanism of tumor progression includes the expression of PD-Ls on tumors, which help facilitate their escape from antitumor immunity by binding to PD-1 on various antitumor effector cells, including CD8^+^ cells. In this study, PD-Ls were found to be expressed in CD14^+^ cells in HCC patients over time as the cancer progressed. We have previously reported the influence of PD-L1^+^PD-L2^+^CD14^+^ cells from patients with advanced stages of HCC on the antitumor activity of CD8^+^ cells. Specifically, PD-L1^+^PD-L2^+^CD14^+^ cells were cocultured with CD8^+^ cells isolated from syngeneic patients, resulting in the antitumor activity of CD8^+^ cells being suppressed against HepG2 cells and Huh7 cells. Meanwhile, their antitumor activity was restored following treatment with a PD-1 antibody, that is, CD14^+^ cells suppressed the antitumor activity of CD8^+^ cells via the PD-L/PD-1 pathway, which can be restored by PD-1 antibodies [19]. Similarly, it has been reported that PD-L1 on dendritic cells mediates CD8^+^ T-cell antitumor activity [31]. In many tumor types, PD-L1 expression is reportedly correlated with PD-1/PD-L1 inhibition. However, patients with very low or absent PD-L1 expression in tumor cells may still derive some benefit from treatment with PD-1 antibody [32]. Hence, the expression of PD-Ls may be responsible for the effect of CD14^+^ cells, which warrants further investigation into the precise associated mechanism.

The expression of PD-L1 is controlled by different mechanisms. PD-L1 constitutive expression in cancer cells may be due to several oncogenic pathways, including chromosome 9 amplification [33,34], PTEN deletions, PI3K/AKT [35], and EGFR mutations [36], MYC overexpression [37], CDK5 disruption [38], and increased PD-L1 transcription [39,40]. The expression of PD-L1 and PD-L2 in tissue Mϕ has been detected in HCC and other cancers [41,42,43]. However, the mechanism of their expression in peripheral blood monocytes remains unclear. Recently, it was reported that the intracellular transfer of cell surface proteins from Reed–Sternberg cells to monocytes, a process known as “trogocytosis”, is induced by direct contact between these cells. Trogocytosis mediates the transfer of PD-L1/L2 from lymphoma cells to monocytes within 1 h [44,45]. Therefore, trogocytosis from cancer cells to monocytes may also occur in HCC patients. However, additional experiments are needed to clarify this issue.

There are certain limitations to note in this study. First, the sample size was relatively small. Notably, this was a longitudinal study, which means repeated blood sampling from the same patients with HCC over a long period of time. Therefore, enrollment was a time-consuming process. Second, patients who had undergone various treatments against HCC were included in the study, and an influence of these treatments on monocyte function cannot be excluded. Therefore, long-term studies with a larger number of patients and more homogeneous cohorts are needed. Third, PD-L1/L2 expression in other antitumor effector cells, such as CD4 T cells, CD8 T cells, B cells, and mast cells, were not examined. However, we have previously reported that the expression of both PD-L1 and PD-L2 in monocytes is associated with poor prognosis [19]. We conclude that monocytes expressing both PD-L1 and PD-L2 may play a key role in antitumor immunity.

## 4. Materials and Methods

### 4.1. Ethics Statement

The study was approved by the Institutional Review Board of the Osaka Medical College. All subjects gave their informed consent for inclusion before they participated in the study. The study was conducted in accordance with the Declaration of Helsinki, and the protocol was approved by the Ethics Committee of Osaka Medical College (approval number: 2125). All animal experiments were carried out in compliance with Japanese regulations. The local institutional animal ethics board of Osaka Medical College approved all mouse experiments (approval number: 26022).

### 4.2. Patients and Specimens

Patients with HCC were classified as “early stage” based on diagnosis of very early- and early-stage HCC and as “advanced stage” based on diagnosis of intermediate- and advanced-stage HCC, according to the Barcelona Clinic Liver Cancer (BCLC) staging system [46]. A total of 168 naïve patients, pathologically confirmed as HCC, were hospitalized in the Osaka Medical College Hospital from April 2010 to January 2018. Fourteen patients with primary or secondary immunodeficiencies (e.g., other cancers, autoimmune diseases, hematologic diseases, infections, chronic heart failures, chronic renal failures, and multiple organ failures) or receiving multikinase inhibitors or immunosuppressive agents were excluded. Among the included patients, 15 patients with early-stage HCC received the initial treatment (radiofrequency ablation) for the purpose of radical cure and cancer progressed to advanced stages after the initial treatment. Blood samples were drawn twice from each patient: at admission for the purpose of initial HCC treatment and after diagnosis of advanced-stage HCC. Clinical characteristics of these patients are shown in Table 1.

Conversely, the study also included five patients with advanced-stage HCC who received the initial treatment that induced tumor regression. Similarly, blood samples were drawn twice from each of these five patients: at admission for the purpose of initial HCC treatment and after diagnosis of early-stage HCC. Clinical characteristics of these patients are shown in Table 2.

### 4.3. Reagents, Media, and Cells

Anti-CD14 magnetic particles-DM and IMag buffer were purchased from BD Biosciences (San Jose, CA, USA). Phycoerythrin-conjugated anti-human PD-L1 monoclonal antibodies (mAbs), allophycocyanin-conjugated anti-human PD-L2 mAbs, IL-12 ELISA MAX kits, and IL-10 ELISA MAX kits were purchased from Biolegend (San Diego, CA, USA). Human rCCL1, rCCL17, and rCXCL13 were purchased from Peprotech (Rocky Hill, NJ, USA). Anti-CCL17 mAbs, anti-CCL1 mAbs, and anti-CXCL13 mAbs were purchased from R&D Systems (Minneapolis, MN, USA). The kits for assessment of cytotoxicity (LDH releasing assay) were from Roche Diagnostics (Mannheim, Germany). HepG2 cells (human hepatoblastoma cells), from DS Pharma Biomedical (Osaka, Japan), were cultured at 37 °C in HepG2 human hepatocellular carcinoma expansion medium (Cellular Engineering Technologies Inc., Coralville, IA, USA). RPMI-1640 medium supplemented with 10% fetal bovine serum was used for CD14^+^ cells.

### 4.4. CD14^+^ Cell Characterization

Ten milliliters of whole blood were drawn into a vacutainer tube containing a small amount of sodium heparin at admission. Peripheral blood mononuclear cells (PBMC) were isolated from heparinized whole blood by Lymphocyte Separation Medium 1077 density gradient centrifugation. PBMC (5 × 10^6^ cells/mL) in IMag buffer were incubated with magnetic beads coated with anti-CD14 mAb (40 min at 4 °C); then, CD14^+^ cells were magnetically harvested. CD14^+^ cells obtained by this procedure were >97% pure, as assessed by flow cytometry [47]. CD14^+^ cells were incubated in fluorescence-activated cell sorting buffer with PE-conjugated anti-human PD-L1, APC-conjugated anti-human PD-L2, or isotype control mAb for 15 min at 4 °C. After washing, PD-L1 and PD-L2 expression was measured using a FACSAria flow cytometer and analyzed by FlowJo software version 10.6.0. In some experiments, CD14^+^ cells (1 × 10^6^ cells/mL) were cultured for 24 h. The culture media were assayed by ELISA for IL-12 (M1Mϕ biomarker), IL-10 (M2Mϕ biomarker), CCL17 (M2aMϕ biomarker), CCL1 (M2bMϕ biomarker), and CXCL13 (M2cMϕ biomarker) [48].

### 4.5. Tumoricidal Activity of CD14^+^ Cells against HepG2 Cell In Vitro

Next, CD14^+^ cells (5 × 10^5^ cells/mL) were stimulated with HepG2 homogenates for 24 h. HepG2 homogenates were made by crushing HepG2 cells (2 × 10^6^ cells/mL) in phosphate-buffered saline with an ultrasonic crusher for 15 min. After washing, CD14^+^ cells were cocultured with HepG2 cells (1 × 10^5^ cells/mL) for 24 h. The tumoricidal activity of CD14^+^ cells against HepG2 cells was measured by LDH release assay [18].

### 4.6. Tumoricidal Activity of CD14^+^ Cells against HepG2 Cells in Humanized Murine Chimeras

Pathogen-free, male NOD.Cg-PrkcscidIl12rgtm1Wjl/SzJ (NSG) mice aged 7–10 weeks were purchased from Jackson Laboratory (Bar harbor, ME, USA). NSG mice lack functional T, B, and NK cells [49,50,51], and their macrophages exhibit defective phagocytosis, digestion, and antigen presentation [52]. The NSG mice were exposed to whole-body X-irradiation (4 Gy) to deplete neutrophils [53] and were defined as xNSG mice in this study. Neutrophils in these animals did not recover for 4 weeks after irradiation. xNSG mice were utilized for the creation of humanized murine chimeras. Specifically, they were inoculated with HepG2 cells in the right groin (2 × 10^6^ cells/mouse). Then, the mice were intravenously inoculated every two weeks with CD14^+^ cells (1 × 10^6^ cells/mouse) isolated from patients with early or advanced-stage HCC. Before inoculation, CD14^+^ cells from early-stage patients were analyzed by flow cytometry for the expression of IL-12, while CD14^+^ cells from advanced-stage patients were analyzed for IL-10 and CCL1 expression. In the chimeras, the inoculated cells spread throughout the body within 2 days of inoculation and were functional for at least 6 weeks. The tumor volume was measured with a microcaliper once a week for 6 weeks and expressed in mm^3^.

### 4.7. Statistical Analyses

Statistical analyses were performed using JMP Pro software version 14 (Tokyo, Japan). Quantitative values are expressed as means. Differences in quantitative values between the two groups were analyzed by Mann–Whitney *U* test. Differences in the ratio of some parameters were analyzed by Fisher’s exact test. Differences with *p* value < 0.05 were considered statistically significant.

## 5. Conclusions

CD14^+^ monocytes from patients with early-stage HCC expressed low levels of PD-L and exhibited antitumor activity. However, CD14^+^ cells from the same patients whose HCC progressed to advanced stages expressed a higher level of PD-L and lacked tumoricidal effects. These findings indicate that the properties of CD14^+^ cells are strongly related to the state of tumor progression in patients with HCC.

## Figures and Tables

**Figure 1 cancers-12-02286-f001:**
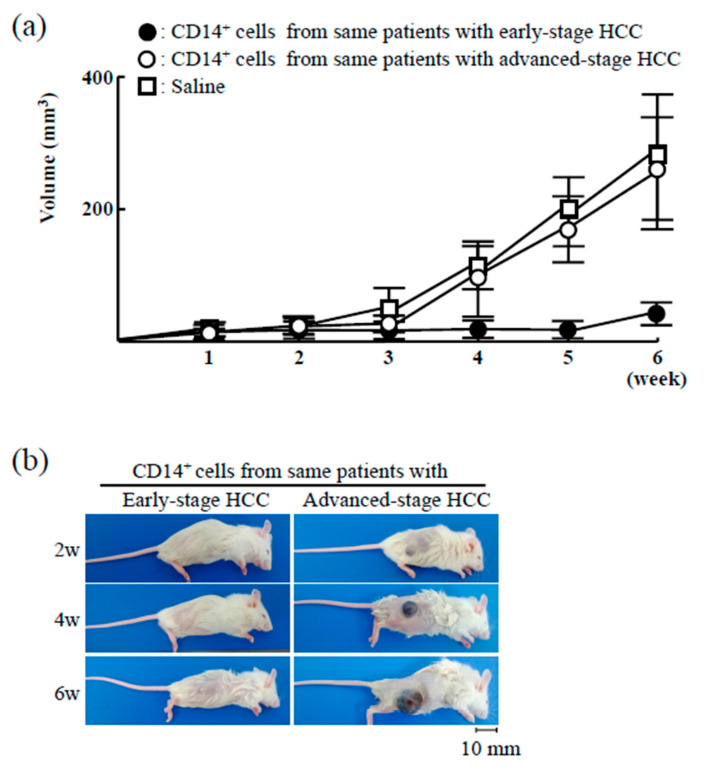
Growth of human HepG2 cells in chimeric mice transplanted with CD14^+^ cells from hepatocellular carcinoma (HCC) patients. (**a**,**b**) Chimeric mice were created in xNSG mice by transplantation of CD14^+^ cells (1 × 10^6^ cells/chimera) from patients with early-stage HCC (closed circles; *n* = 5). CD14^+^ cells were also isolated from those patients who were later diagnosed with advanced-stage HCC, and new chimeric mice were created from xNSG mice by the same method (open circles; *n* = 5). Both groups of chimeric mice were subcutaneously inoculated with HepG2 cells (2 × 10^6^ cells/mice). xNSG control mice received saline along with tumor cell inoculation (open squares; *n* = 3). Tumor size was measured with a microcaliper, and tumor volume is expressed in mm^3^.

**Figure 2 cancers-12-02286-f002:**
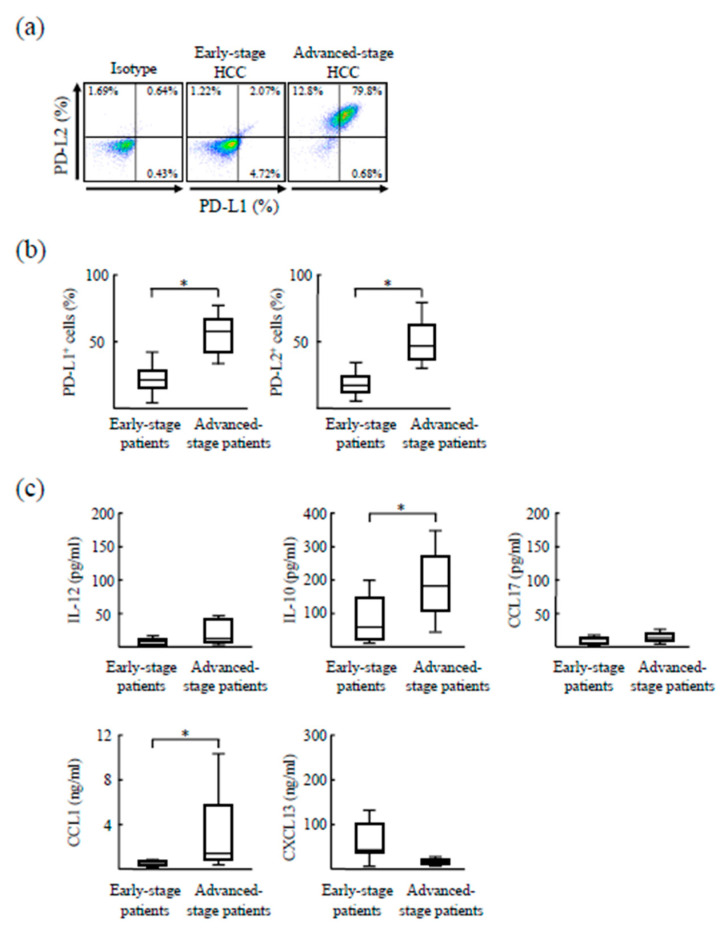
Differences in the properties of CD14^+^ cells in the same patients at early and advanced HCC stages. (**a**,**b**) CD14^+^ cells were isolated from HCC patients (*n* = 12) at early tumor stages before initial treatment and at advanced stages after treatment initiation. These cells were stained with anti-PD-L1 and anti-PD-L2 antibodies and assayed by flow cytometry. (**c**) CD14^+^ cells were cultured for 24 h, and the culture media was assayed by ELISA for IL-12, IL-10, CCL17, CCL1, and CXCL13 production. * *p* < 0.05. PD-L, programmed death ligand.

**Figure 3 cancers-12-02286-f003:**
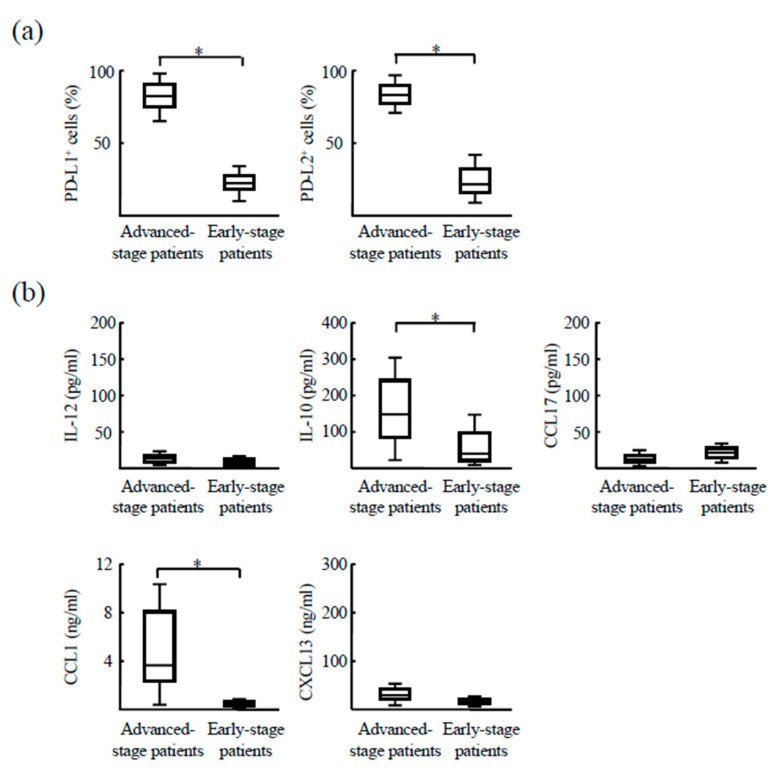
Changes in CD14^+^ cell properties related to treatment-induced tumor regression in patients with advanced-stage HCC. (**a**) CD14^+^ cells were isolated from patients with advanced HCC before treatment initiation and after tumor regression (*n* = 5). CD14^+^ cells were stained with anti-PD-L1 and anti-PD-L2 antibodies and assayed by flow cytometry. (**b**) CD14^+^ cells were cultured for 24 h, and their culture media was analyzed by ELISA for IL-12, IL-10, CCL17, CCL1, and CXCL13 detection. * *p* < 0.05.

**Figure 4 cancers-12-02286-f004:**
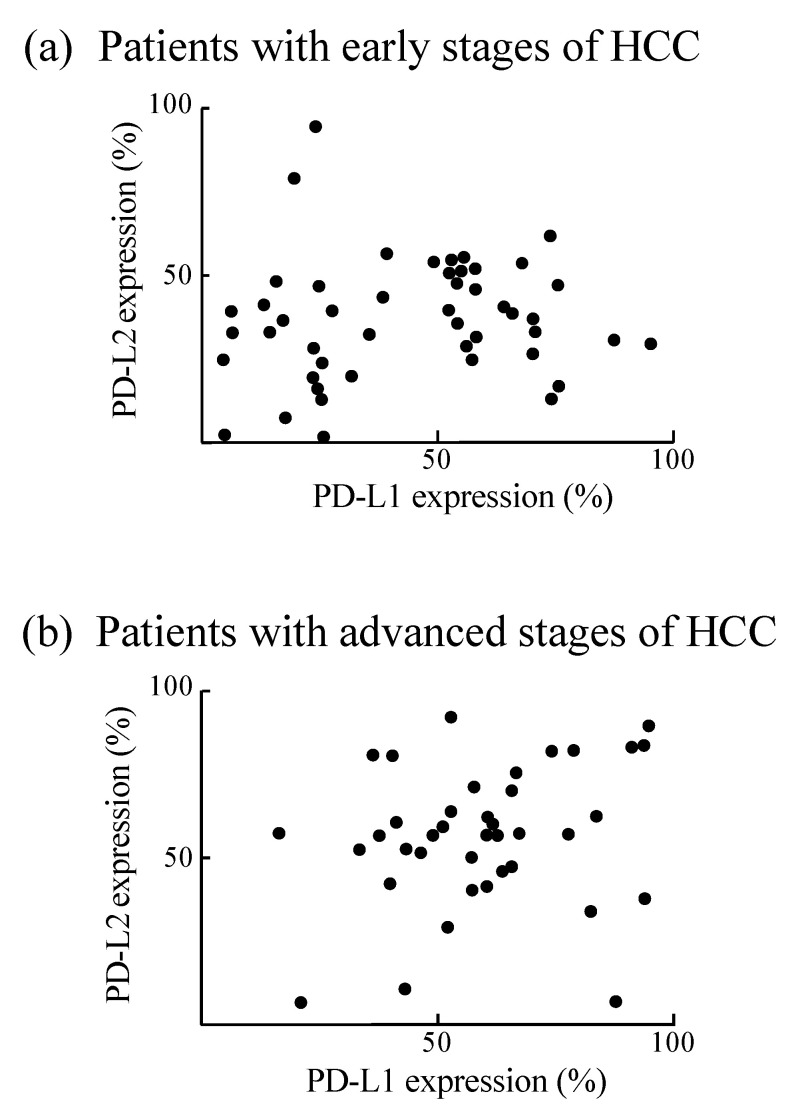
PD-L expressions in CD14^+^ cells from patients with several stages of HCC. CD14^+^ cells were isolated from peripheral blood of patients with early stages of HCC (*n* = 48) (**a**) and advanced stages (*n* = 41) (**b**). These cells were stained for PD-L1 and PD-L2 and assayed for flow cytometry.

**Figure 5 cancers-12-02286-f005:**
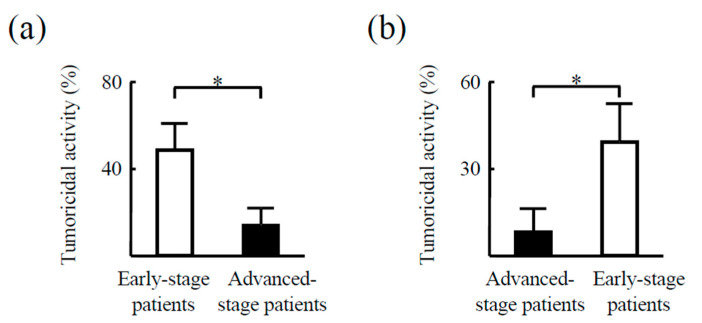
Tumoricidal activity of CD14^+^ cells depends on cancer progression. CD14^+^ cells isolated from patients before and after tumor regression were examined for cytotoxic activity against HepG2 cells by lactate dehydrogenase (LDH) release assay. (**a**) Changes related to tumor progression from early stages to advanced stages (*n* = 5). (**b**) Changes related to tumor progression from advanced stages to early stages (*n* = 5). * *p* < 0.05.

**Table 1 cancers-12-02286-t001:** Changes in the clinical characteristics of the same patients with early- and advanced-stage HCC.

(*n* = 15)	Early Stages	Advanced Stages	*p*-Value
Age (year, range)	72.4 (62–82)		
Gender (Male/Female)	14/1		
Etiology (%)			
HBV	2 (6.7)		
HCV	5 (33.3)		
Others	8 (53.3)		
Child-Pugh class			
A/B/C	15/0/0	15/0/0	
TNM stage			<0.001
I/II/III/IV	3/12/0/0	0/0/14/1	
WBC (×10^6^ /mL, mean ± SD)	4.69 ± 1.23	4.64 ± 2.01	0.948
Neutrophils (×10^6^ /mL, mean ± SD)	2.47 ± 0.87	2.69 ± 1.40	0.794
Lymphocytes (×10^6^ /mL, mean ± SD)	1.45 ± 0.85	1.35 ± 0.73	0.788
Monocytes (×10^5^ /mL, mean ± SD)	3.58 ± 1.02	3.73 ± 1.40	0.796
Platelets (×10^4^ /mL, mean ± SD)	12.8 ± 6.0	10.5 ± 4.5	0.483
AST (IU/L, mean ± SD)	56.5 ± 28.2	54.5 ± 27.2	0.873
ALT (IU/L, mean ± SD)	47.8 ± 28.7	38.5 ± 21.3	0.417
Albumin (g/dL, mean ± SD)	3.55 ± 0.47	3.53 ± 0.53	0.928
Total bilirubin (mg/dL, mean ± SD)	0.98 ± 0.56	1.0 ± 0.87	0.935
Prothrombin time (%, mean ± SD)	87.0 ± 13.5	91.2 ± 18.7	0.360
CRP (mg/dL, mean ± SD)	0.35 ± 0.53	0.17 ± 0.15	0.367
AFP (ng/mL, range)	50.8 (3.7–237.0)	63.1 (4.5–185.0)	0.713
DCP (mAU/mL, range)	252.4 (8.0–1750)	378.8 (14.1–2910)	0.691

HBV, hepatitis B virus; HCV, hepatitis C virus; WBC, white blood cells; SD, standard deviation; AST, aspartate aminotransferase; ALT, alanine aminotransferase; CRP, C-reactive protein; AFP, alpha fetoprotein; DCP, des-gamma-carboxyl prothrombin.

**Table 2 cancers-12-02286-t002:** Changes in the clinical characteristics of same patients following tumor regression.

(*n* = 5)	Advanced Stages	Early Stages	*p*-Value
Age (year, range)	74.4 (61–80)		
Gender (Male/Female)	4/1		
Etiology (%)			
HBV	0 (0.0)		
HCV	4 (80.0)		
Others	1 (20.0)		
Child-Pugh class			
A/B/C	5/0/0	5/0/0	
TNM stage			0.128
I/II/III/IV	0/1/4/0	2/3/0/0	
WBC (×10^6^ /mL, mean ± SD)	4.87 ± 1.10	4.63 ± 1.09	0.764
Neutrophils (×10^6^ /mL, mean ± SD)	3.01 ± 0.69	2.79 ± 1.20	0.762
Lymphocytes (×10^6^ /mL, mean ± SD)	1.28 ± 0.40	1.25 ± 0.54	0.939
Monocytes (×10^5^ /mL, mean ± SD)	3.01 ± 0.51	2.89 ± 1.24	0.861
Platelets (×10^4^ /mL, mean ± SD)	15.5 ± 5.5	14.3 ± 4.6	0.483
AST (IU/L, mean ± SD)	28.4 ± 11.9	29.2 ± 14.1	0.873
ALT (IU/L, mean ± SD)	25.2 ± 16.5	23.6 ± 14.0	0.417
Albumin (g/dL, mean ± SD)	4.26 ± 0.29	4.0 ± 0.34	0.350
Total bilirubin (mg/dL, mean ± SD)	0.72 ± 0.31	0.76 ± 0.34	0.935
Prothrombin time (%, mean ± SD)	86.2 ± 8.7	72.8 ± 10.3	0.360
CRP (mg/dL, mean ± SD)	0.12 ± 0.12	0.19 ± 0.14	0.41
AFP (ng/mL, range)	50.8 (2.6–171.3)	8.3 (2.3–21.5)	0.933
DCP (mAU/mL, range)	203.7 (52.0–809)	65.9 (21.7–115.9)	0.57

HBV, hepatitis B virus; HCV, hepatitis C virus; WBC, white blood cells; SD, standard deviation; AST, aspartate aminotransferase; ALT, alanine aminotransferase; CRP, C-reactive protein; AFP, alpha fetoprotein; DCP, des-gamma-carboxyl prothrombin.

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
