# Peer review of "Programmed Death 1 Ligand Expression in the Monocytes of Patients with Hepatocellular Carcinoma Depends on Tumor Progression"

_cancers, 2020, doi:10.3390/cancers12082286_

Round 1

Reviewer 1 Report

Manuscript entitled “Program Death 1 Ligand Expression in the Monocytes of Patients With Hepatocellular Carcinoma Depends on Tumour Progression” is a straight forward manuscript explaining the role of PD-L in HCC progression and its association with CD14+ cells. The authors have isolated CD14+ cells from early and advance stages of HCC patients and inoculated in humanized murine chimera with HepG2 cells to see their effects in HCC progression.   CD14+ cells from patients at early HCC stages exhibited antitumor activity and expressed low level of PD-L while CD14+ cells from advance stages of HCC lacked antitumor activity and expressed high levels of PD-L. The authors correlate CD14+ cells properties with HCC progression and tumour stages. Overall the manuscript looks interesting, however I have few concerns.

- Abstract seems very confusing at first read. It should be re-structured and results in abstract should be described in sequence to avoid confusions for the readers.

- Few lines describing impact of present study on the field of oncology can also be added at the end of the abstract.

- Author should need to highlight some more information regarding the association among CD14+ and PD-L, IL-10 and CCL1 production in introduction part. For example, there are largest types of monocytes and why the authors just choose CD14+ to prove the association with cancer progression. Do the authors have any other previous results to prove the significance of CD14+ rather than other types of monocyte?

- I am just wondering if the authors have checked the expression levels of PD-L among early stage patients to see if there is some variation in the expression levels of PD-L among patients themselves.

Same goes with advance stage patients as well.  

Some in vitro experiments can be added to address the above mentioned concerns.

- When the authors described the experiment, I believe more details should be included.  For example: In the animal experiment, how the authors inject those HepG2 cells and CD14+ cells?

- In results authors have concluded that CD14+ cells from patients with early-stage HCC exhibited poor PD-Ls, IL-12, and IL-10 expressions, and were negative for CCL17, CCL1, and CXCL13 cells (considered a quiescent MФ phenotype). On the other hand, CD14+ cells from the same patients at advanced HCC stages had acquired PD-L1 and PD-L2 expressions and exhibited an IL-12-IL10+CCL17-CCL1+CXCL13- phenotype (considered an M2bMФ phenotype)”. These phenomena can be discussed in detail and possible mechanisms can also be explained in discussion part. In addition, for the mice model, Fig 1 was observed the Growth of HepG2 cell-derived tumors in chimeric mice with CD14+ cells from HCC patients. Why the authors didn’t perform the animal study by using the CD14+ cells from the same patients to recover tumoricidal activity?

- I am also wondering whether they kept the isolated CD14+ cells for an year from early stage HCC patients and after isolating CD14+ cells from advanced patients, inoculated in humanized murine chimera or two separate experiments were performed at different time point ????? This should be specify in the methodology part as age of specimens may lead to an underestimation of PD-L1 status.

-I suggest authors to add some in vitro experiments to validate their findings and hypothesis at the same time as the sample size in the study seems inadequate.

- Discussion part should be more explanatory and some more evidence supporting the current results should be cited since the number of experiment as well as sample size are inadequate. In general, I think the data presented in the manuscript are not enough for the publication in Cancers and the hypothesis should be look into much deeper. For example, in the authors' discussion section, the authors mentioned CD14+ may be one of the reasons why immune checkpoint inhibitors are effective in patients with PD-L1 negative tumours. Thus, I believe more studies (both in vitro and in vivo) need to be performed to prove the this 'claim'.

- Finally, the manuscript should also be improved in term of English language. Misuse of punctuation can also be seen at several places in manuscript.

Reviewer 2 Report

Overall comments:

This manuscript deals with the relationship between disease progression and PD-L expression in monocytes of patients with HCC; several questions remain open, notably on the influence of therapy; patients who had undergone various treatments against HCC were included in the study – see last paragraph of the discussion.

Specific comments:

Please check the numbers and units (e.g., "platelets: 12.8 ± 6.0 x10^4/ml"? "albumin: 3.55 ± 0.47 mg/dL"? etc.)

Reference list: consistent use of journal abbreviations is recommended.

Round 2

Reviewer 1 Report

The revised manuscript has addressed most of my previous concerns/questions. However, there are still few minor points need to be fixed before the manuscript can further consider for publication.

  • For better clarification, I believe the authors should also named "program death 1 ligand" as "PD-L / PD-1" [not just only as "PD-L"]
  • Page 2 Line 48 -- Tumor-associated macrophages should be as "TAMs"  
  • What is M1 and M2? Please clarify
  • I believe the Conclusions section should be stright after the Discussion section
  • Supplementary Figure should also be included as one of the Fgiures. However, more explanatory/descriptive are reuqired for this experimental work within the contents
  • Some recent/latest publications related in this research area should be considered to use as reference(s) -- For examples:Guo et al. Frontiers in Immunology 11:1508 · July 2020Jilkova et al. Cancers (Basel) 11(10): 1554 Oct 2019
